



**1  Drought reconstruction since 1796 CE based on tree-ring widths in the Upper**

**2  Heilongjiang (Amur) River Basin in Northeast Asia, and its linkage to Pacific**

**3  Ocean climate variability**

Yang Xu [1], Heli Zhang [2], Feng Chen [1]*, Shijie Wang [1], Mao Hu [1], Martín Hadad [3], Fidel Roig [4, 5]
*1. Yunnan Key Laboratory of International Rivers and Transboundary Eco-Security, Institute of*
*International Rivers and Eco-Security, Yunnan University, Kunming 650500, China*
*2. Key Laboratory of Tree-ring Physical and Chemical Research of China Meteorology*
*Administration/ Xinjiang Key Laboratory of Tree-ring Ecology, Institute of Desert Meteorology,*
*China Meteorological Administration, Urumqi 830002, China*
*3. Laboratorio de Dendrocronología de Zonas Áridas. CIGEOBIO (CONICET-UNSJ), San Juan,*
*Argentina, Gabinete de Geología Ambiental (INGEO-UNSJ), Av. Ignacio de la Roza 590 (oeste),*
*J5402DCS Rivadavia, San Juan, Argentina*
*4. Laboratorio de Dendrocronología e Historia Ambiental, IANIGLA-CCT CONICET, Mendoza,*
*Argentina*
*5. Hémera Centro de Observación de la Tierra, Escuela de Ingeniería Forestal, Facultad de*
*Ciencias, Universidad Mayor, Camino La Pirámide 5750, Huechuraba, Santiago 8580745, Chile*
*Correspondence: feng653@163.com
**Abstract:** The economic and environmental impacts of persistent droughts in East
Asia are of growing concern, and therefore it is important to study the cyclicity and
causes of these regional droughts. The self-calibrating Palmer Drought Severity Index
(scPDSI) has been extensively employed to describe the severity of regional drought,
and several PDSI reconstructions based on tree rings have been produced. We
compiled a tree-ring chronology for Hailar pine *(Pinus sylvestris* var. *Mongolica)*
from two sites in the Hailar region in the Upper Heilongjiang (Amur) River Basin.
Analysis of the climate response revealed that scPDSI was the primary factor limiting

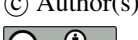



tree ring growth from May to July. The mean May to July scPDSI in the Hailar region
since 1796 was reconstructed from the tree-ring width chronology. The results of
spatial correlation analysis revealed that the reconstructed scPDSI in this region
responded significantly   to climate change. Analysis of the synoptic climatology
indicated that the drought in the Upper Heilongjiang (Amur) River Basin is closely
related to ENSO and the Silk Road teleconnection. The results of atmospheric water
cycle analysis show that water vapor transport processes are the dominant factor in
the development of drought in this region.
**Keywords: Tree rings; ScPDSI reconstruction; Sea surface temperature;**
**Severe drought; Moisture recycling**
## 1.   Introduction
Drought—accompanied by persistent high temperatures and below-average
precipitation over intervals of months to years—is of growing concern. As a natural
disaster, the frequency and duration of drought have increased as global warming has
intensified. The impact of drought on human well-being and economic productivity is
also increasing, given that drought severely threatens food and water security (Lesk et
al. 2016; Trenberth et al. 2014; Wang et al. 2016; Chen et al. 2022). Due to regional
water shortages, droughts frequently wreak havoc on agriculture and the quality of
life in northeast China (NEC). Hence, understanding the variability of drought in this
region and its causal mechanisms is essential for both drought prediction and the
formulation of disaster response strategies (Li et al. 2019; Yuan and Wood 2013).
However, only short-duration instrumental records of drought variability are





available for NEC, most of them from the 1950s onwards. However, this deficiency
can be addressed via proxy paleoclimate records, such as tree-ring widths (Fritts,
1991). With their high annual precision and extensive coverage, tree rings have been
used as a reliable proxy for reconstructing historical climatic and hydrological
changes (Cook et al. 2016; Chen et al. 2021; Pearson et al. 2020). Hailar is located in
the Upper Heilongjiang (Amur) River Basin, in the woodland-steppe interface of NEC,
part of the eastern edge of the Hulunbuir grasslands, a region highly susceptible to
climatic and environmental changes and that has experienced drought over the past
few decades (Zhang et al. 1997; Wang et al. 2010; Bao et al. 2015; Chen et al. 2012).
Drought reconstructions based on tree-ring widths can potentially make a valuable
contribution to regional planning and ecological conservation in this region. Over the
past two decades, several studies based on tree-ring width have been conducted in
Northeast Asia (Cook et al. 2010; Liang et al. 2007; Bao et al. 2015; Chen et al. 2012;
Liu et al. 2016; Liu et al. 2009). However, research attention needs to be directed to
the agro-pastoral zone in the western part of NEC, where the fragile ecology and
climate sensitivity necessitate a greater understanding of the patterns and mechanisms
of drought.

Severe drought events are a serious problem in northern China, especially since the

late 1970s, when the weakening of the East Asian Summer Monsoon (EASM)
contributed to the 'southern flooding and northern drought' climatic pattern, with
frequent intense drought events in the north (Wang, 2002; Yu et al. 2004; Ding et al.
2009). Regarding the climatic mechanisms responsible for the NEC drought, it has



been suggested that variations in the Pacific Ocean interdecadal oscillation (PDO) and
in   Arctic Ocean sea-ice cover have contributed to an interdecadal decrease in
precipitation in NEC, leading to drought (Han et al. 2015). It has also been suggested
that the global distribution of sea surface temperature and ENSO events are closely
linked to summer precipitation in NEC, thus explaining the summer drought
mechanism in the NEC from an interannual perspective (Han et al. 2017). Winter
NAO has also been shown to impact the interannual variability of summer drought
events in NEC (Fu and Zeng, 2005). Anticyclonic circulation anomalies can often
trigger extreme and prolonged drought events. Such anomalies always occur as a
major product of specific remote teleconnection patterns, called stationary wave
patterns (Schubert et al. 2014). Several steady wave models have been shown to
generate extreme drought events, with the 2014 summer drought in northern China
attributed to the EU pattern. It has also been confirmed that the Silk Road,
Pacific-Japanese, and EU models caused the July–August 2014 drought in north and
northeastern China (Wang and He, 2015; Wang et al. 2017; Xu et al. 2017). While
many of the above studies describe water vapor flux anomalies during periods of
extreme drought, our understanding of the role of water vapor derived from local
evaporation and advective transport is limited. Quantifying the contribution of
advected water vapor transport and precipitation circulation processes to precipitation
is essential for understanding the water vapor cycle and anticipating the intensity of
severe drought episodes (Findell and Eltahir, 2003; Guan et al. 2022).

The objectives of the present study are: (1) To reconstruct the scPDSI of the Hailar



region and to analyze changes in the temporal variations of regional drought; (2) to
determine the atmospheric circulation mechanisms generating extreme drought events;
and (3) to analyze the contribution of advective water vapor transport and local
evaporation to precipitation during droughts, and to determine their leading causes.
**2. Materials and Methods**
**2.1 Study area**
Tree-ring sampling sites NEGC (119°36′ E, 47°58′ N, 600-700m a.s.l.) and MGET
(119°24′ E, 47°59′ N, 1100-1200 m a.s.l.) are located in the Upper Heilongjiang
(Amur) River Basin (Fig. 1). The region lies within the arid and semi-arid region of
NEC, on the eastern edge of the Hulunbeier steppe and close to the western slopes of
the Greater Khingan Range. This region has a continental and monsoonal climate.
Due to the incursion of high-latitude cold and dry air masses in winter and of warm
and moist air masses from low-latitude areas in summer, the climate tends to alternate
between cold and dry in winter and warm and humid in summer. The average annual
temperature is around -0.9 °C and the average yearly precipitation is ~382.8 mm (Fig.
2a). December–January is the coldest period, with sparse rainfall, while June–August
is the hottest period when precipitation is abundant (Fig. 2b). Thus, the climate is
generally cold and dry. The grassland in this region is undergoing severe
desertification and degradation in response to global and regional climate change
(Zhang et al. 2011).
**2.2 Tree-ring data**
The dominant tree species in the Hailar region is Hailar pine (*Pinus sylvestris* var.



*Mongolica*), which was sampled for tree-ring analysis. Both sites were located at the
upper tree line, on steep slopes with thin soils. Information about the sampling sites is
given in Table 1. Samples were taken from chest height using a 10-mm diameter
incremental borer. Forty cores were collected from 20 trees at sampling site NEGC,
and 63 cores were collected from 33 trees at sampling site MGET. In the laboratory,
the core samples were dried, mounted and successively sanded with 320- and 600-grit
sandpaper until the tree-ring widths were visible, and were then imaged using a
high-precision scanner. Tree-ring width data were measured using CooRecorder 9.4
software, and the data quality was checked by cross-matching using the quality
control program COFFCHA (Holmes, 1983). The ARSTAN procedure was then used
to remove non-climatic influences on the tree-ring width data, due to age and growth,
using      exponential detrending. This procedure resulted in a standardized
chronology of tree-ring widths (STD), a chronology of differences (RES), and an
autoregressive chronology (ARS). The individual detrended chronologies from the
two sites were combined to produce a new RC chronology using a robust averaging
method (Cook, 1985). The STD chronology was selected to retain high and
low-frequency variations based on the considerations of subsequent analyses. The
data series were truncated according to thresholds of at least EPS > 0.85 and 6 (3 trees)
for the expressed population signal and sample size, respectively, resulting in a
reliable reconstruction for the period of 1796–2020.
**2.3 Climate data and statistical methods**
Monthly instrumental climate data from Hailar meteorological station (49°15′ E,



119°42′ N, 650 m a.s.l.), affiliated to the National Meteorological Administration of
China, including monthly mean temperature and monthly total precipitation, were
obtained for the period of 1951–2020. Monthly mean runoff data from Khabarovsks
Hydrological Station on the lower Heilongjiang River were used to analyze the
response of the reconstructed scPDSI to runoff variations. The locations of the
meteorological and hydrological stations are shown in Fig. 2a. scPDSI gridded
climate data of CRU TS 4.06 from the Climate Research Unit (CRU) of the
University of East Anglia were also used in this study (Harris et al. 2014). SPSS 22.0
was used to assess the correlation of the climate signals contained in the three
chronologies for the individual months from July of the previous year to September of
the current year. Based on the results of this correlation analysis, several seasonal
climate combinations from July of the last year to September of the current year were
filtered, and the seasonal climate combinations with the highest correlation were
selected for climate reconstruction, using one-dimensional linear regression. A split
calibration-verification test was used to test the reliability of the reconstructed models,
dividing the period of 1951–2020 into independent calibration and validation periods.
The main parameters assessed were the correlation coefficient (R), explained variance
($R^2$), efficiency coefficient (CE), error reduction value (RE), sign test (ST1), and the
first-order difference sign test (ST2) (Cook and Kairiukstis, 2013). In this study, after
15-year low-pass filtering, intervals of more than 10 years below/above the mean of
the reconstructed series were defined as    dry/wet periods, and the years below or
above 1.5 times the standard deviation of the series mean were defined as extreme



dry/wet years. The quasi-periodic characteristics of the reconstructed scPDSI were
analyzed using Multitaper spectral analysis (MTM) (Mann and Lees, 1996). Spatial
correlation maps were generated between the reconstructed scPDSI series and the grid
data, including precipitation and scPDSI data from CRU TS 4.06, and runoff grid
point data from G-RUN (Harris et al. 2014; Ghiggi et al. 2021).

### 2.4 Land-atmosphere water balance

The Brubaker binary model has been used to quantify the contribution of external
water vapor transport and local evaporative water vapor to precipitation, based on the
atmospheric water vapor balance (Brubaker et al. 1993). The water vapor equation for
the vertical integration per unit area can be expressed as follows (Brubaker et al. 1993;
Guo et al. 2018):

$$\frac{\partial Q}{\partial t} = -\left(\frac{\partial F_u}{\partial x} + \frac{\partial F_v}{\partial y}\right) + E - P, \tag{1}$$

Where Q is the vertically integrated water vapor concentration; $F_u$ and $F_v$ are the
vertically integrated latitudinal and meridional water vapor fluxes, respectively; and
E and P are the vertically integrated land evaporation and rainfall, respectively.
Compared to the magnitude of the water vapor flux, the vertically integrated water
vapor content varies very little over time and is insignificant on longer timescales
(Burde and Zangvil, 2001). Thus, if the left side of equation (1) is 0, we obtain the
following equation:

$$\left(\frac{\partial F_u}{\partial x} + \frac{\partial F_v}{\partial y}\right) = E - P, \tag{2}$$

Assuming that externally imported water vapor and locally evaporated water vapor
are well mixed over the study area, and that the proportions of evaporated and



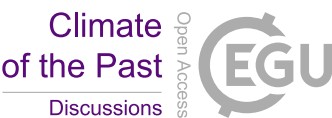

advected water vapor contribute equally to the development of precipitation and
moisture fluxes, the proportional relationship can be obtained, as follows (Zhao and
Zhou, 2021; Guo et al. 2018; Li et al. 2020):

$$\frac{P_a}{P} = \frac{Q_a}{Q}, \tag{3}$$

$$\frac{P_e}{P} = \frac{Q_e}{Q}, \tag{4}$$

Where $Q_a$ and $Q_e$ represent the water vapour content resulting from external
water vapour transport and local land surface evaporation, respectively, and $P_e$ and
$P_a$ are the precipitation amounts resulting from evaporation and the external transport
of water vapor, respectively. In addition, the water vapor balance equation for the
external water vapor transport term is as follows (Guo et al. 2018; Zhao and Zhou,
2021; Li et al. 2020):

$$-\left(\frac{\partial F_u^a}{\partial x} + \frac{\partial F_v^a}{\partial y}\right) = P_a \tag{5}$$

Where $F_u^a$ and $F_v^a$ represent the vertically integrated latitudinal and longitudinal
water vapor transport from external inputs, respectively, assuming P, E and $P_a$ are
constant within the study area during the interval of concern (Burde and Zangvil,
2001). Using the above assumptions and the Gaussian scattering assumptions,
equations (2) and (5) can be applied to a region of area A (in m), as follows:

$$-\left(\frac{\partial F_u}{\partial x} + \frac{\partial F_v}{\partial y}\right)|A = F_{in} - F_{out} = (P - E)A \tag{6}$$

$$-\left(\frac{\partial F_u^a}{\partial x} + \frac{\partial F_v^a}{\partial y}\right)|A = F_{in} - F_{out-a} = P_a A \tag{7}$$

Here, $-\left(\frac{\partial F_u}{\partial x} + \frac{\partial F_v}{\partial y}\right)|A$ and $-\left(\frac{\partial F_u^a}{\partial x} + \frac{\partial F_v^a}{\partial y}\right)|A$ represent the total water vapour
irradiation dispersion in the targeted region and the irradiation dispersion of externally
transported water vapor, respectively; $F_{out}$ and $F_{out-a}$ represent the total water



vapour leaving the calculated area and the part of the external input water vapour
flowing away from the calculated area again, respectively; and $F_{in}$ represents the
total water vapor transported to the targeted area from outside. This enables an
estimate to be made of the contribution of external moisture transport and local land
surface evaporation to precipitation, as follows (Guo et al. 2018; Li et al. 2020):

$$r = \frac{P_a}{P} = \frac{2F_{in}}{2F_{in} + EA} \tag{8}$$

$$\rho = 1 - \frac{P_a}{P} = \frac{EA}{2F_{in} + EA} \tag{9}$$

Where $r$ and $\rho$ are the contributions to precipitation from external water vapor
transport and local land surface evaporation, respectively, and $\rho$ is the precipitation
recirculation rate.
The Brubaker binary model water vapor transport process is based mainly on
advection terms, which can be applied to calculate the precipitation recirculation rates
in the study area. Give that the calculation of these precipitation recirculation rates
depends on the size of the selected area, the study area was enlarged (42.5–52.5°
N,115–125° E) for the purpose of calculation.
**3. Results**
**3.1 scPDSI reconstruction**
All the tree ring chronologies show a high mean sensitivity and standard
deviation, typical of trees growing in arid and semi-arid regions, due to the location of
the Hailar region. The high inter-series correlation suggests that our tree-ring width
chronology reliably captures several standard climate signals. The EPS of the RC
chronology passed the test for signal strength (EPS > 0.85) after 1796 (Table 2 and





Fig. 3). The tree-ring width series has a significant negative correlation with
temperature, a significant positive correlation with precipitation, and a significant
positive correlation with scPDSI, according to the climate response results ($p < 0.05$)
(Fig. 4a, b). Screening for seasonal combinations of temperature, precipitation, and
scPDSI revealed the strongest correlation between the RC tree ring width chronology
and meant scPDSI from May to July (r = 0.645, p < 0.01). Accordingly, we
reconstructed the May to July scPDSI for the Hailar region since 1796 CE, using the
following equation (Fig. 4d):

$$Y = 3.681X - 4.146 \qquad\qquad (10)$$

$$(n = 70,\ r = 0.645, R^2 = 41.6\%, R^2_{adj} = 40.7\%, F = 48.385, p < 0.01)$$

Where $Y$ is the mean reconstructed scPDSI for May to July, and $X$ is the tree ring
width index from the composite chronology.
In equation (10), the correlation between the mean May–July scPDSI and the
tree-ring width index over the period of 1951–2020 is 0.645, with the tree-ring width
index explaining 41.6% (40.7% after adjustment for the degrees of freedom) of the
mean scPDSI variance, $F = 48.385$ and $p < 0.01$. Except for several anomalously high
values, the reconstructed mean scPDSI values agree well with the instrumental data
(Fig. 4c). The split calibration-verification test results show that the reconstruction
model has good reliability and stability, with values of RE and CE > 0.20. The sign
and first-order difference sign tests are significant at the 0.05 level (Table 3). These
results suggest that our scPDSI reconstruction has reliably recorded climate signals at
low frequencies.



## 3.2 Characteristics of the scPDSI reconstruction

Our scPDSI reconstructions reveal oscillations between drier and wetter conditions in the Hailar region during 1796–2020 CE (Fig. 4e). Dry/wet periods after 15-year low-pass filtering were continuously below/above the long-term mean for more than 10 years. Four dry periods (1809–1819, 1829–1878, 1937–1950, 1990–2012), and five wet periods (1796-1808, 1879-1900, 1910-1936, 1951-1963, 1970-1989) are evident in the record. A data value < 1.5 times the standard deviation of the long-term mean is defined as an extreme drought year, and such years occurred in 1779, 1826, 1837, 1840, 1842, 1857, 1864, 1866, 1951, 1996 and 2007. The curves also show an increase following   lower values in the 1870s, and a clear decreasing trend in the last 10 years, which is consistent with the instrumental observations (Fig. 4e). The results of the MTM analysis revealed periodicities of 2–8.1 years (Fig. 5). The results of spatial correlation analysis revealed a strong positive correlation between the reconstructed scPDSI series on the scale of the upper basin of the Heilongjiang (Amur) River and the gridded scPDSI, total rainfall, and runoff, from May to July (Fig. 6a, b). After obtaining the mean series of the gridded data, good correlations were obtained between the reconstructed scPDSI and the regional mean of the gridded data, with r = 0.57 (p < 0.01), and r = 0.35 (p < 0.01), with CRU scPDSI and CRU precipitation, respectively (Fig. 6a, b, c). The correlations between reconstructed scPDSI and G-RUN runoff and runoff from the Khabarovsks Hydrological Station runoff were r = 0.34 (p < 0.01) and r = 0.36 (p < 0.01), respectively (Fig. 6d). These results indicate that our scPDSI reconstructions reliably reflect the regional drought characteristics



and changes in runoff in the Upper Heilongjiang (Amur) River Basin.

## 4. Discussion

### 4.1 Climate–tree ring growth relationships and temporal variations in regional drought

The positive correlation between tree-ring width and rainfall and the negative
correlation with temperature indicate that the increase in the circumference of *Pinus*
*sylvestris* var. *Mongolica* in the Hailar area is described by a humidity-sensitive
growth model. Temperature is much a greater stressor for tree growth in arid and
semi-arid regions than precipitation (Bao et al. 2015; Fang et al. 2010; Sun et al.
2012). The higher correlation coefficients between temperature and the tree-ring
indices in our dataset    indicate that the radial expansion of *P. sylvestris* var.
*Mongolica* in the Hailar region is    mainly influenced by soil moisture conditions
modulated by temperature variations (Fig. 4a). Compared with precipitation alone,
PDSI better reflects changes in soil moisture caused by precipitation and temperature
stress on the radial growth of trees. The PDSI during the growing season from May to
July also shows the highest correlation with scPDSI ($r = 0.645$, $p < 0.01$) (Fig. 4c).
The radial growth of *P. sylvestris* var. *Mongolica* is mainly determined by the control
of soil moisture by precipitation (Song et al. 2015). However, in semiarid areas, the
increasing temperature during the growing season    accelerates the evaporation of soil
moisture and enhances plant transpiration, and thus the soil moisture supply is
insufficient for tree growth (Shang et al. 2012). In contrast, temperatures above a
certain threshold during the growth season can adversely affect tree growth because



the decrease in the net photosynthetic rate and excessive temperatures will lead to
more severe drought stress (D'arrigo et al. 2004).
The reconstructed scPDSI reveals ten extreme drought years during 1796–2000,
seven of which can be identified in historical documents (Zhang, 2004; Liu and Wen,
2008). (Table 4). The historical literature includes detailed descriptions of drought
events; for example, 1951 was a drought year throughout Inner Mongolia—one of a
series of relatively severe droughts—when the lack of rainfall in summer and autumn
was more severe than in spring. Numerous seedlings of crop plants in Hulunbuir were
killed by the drought and the grain yield of the entire region was significantly reduced
(Liu and Wen, 2008). In 1996, a severe drought affected the north-central part of Inner
Mongolia in early summer (Liu and Wen, 2008). Our reconstruction captures several
extreme drought events in the past decade. The intense heat in NEC during
July–August 2016 resulted in severe crop yield reductions and economic losses
amounting to \$15,61 billion (Li et al. 2018). In 2017, NEC experienced the most
severe spring and summer drought event of the last few decades (Zeng et al. 2019),
which heavily affected the cultivated area in eastern Inner Mongolia, the magnitude of
the crop failure and direct economic losses were the second highest since 2012, with
the area of $74.3 \times 10^4$ km$^2$ being affected by drought across the region, and with
moderately intense drought occurring mainly in western Hulunbuir (Zhang et al.
2017). NEC is a major food-producing region in China, and thus it is of both regional
and national importance to improve our understanding of the causes and patterns of
drought events and to develop appropriate responses.



## 4.2 Synoptic meteorological analysis of severe drought


To explore the climatic drivers of the extreme drought events, we screened the
wettest and driest decades from 1891 to 2020. SST changes in the previous winter are
critical for precipitation in East Asia in the following year (Juneng and Tangang,
2005), and thus we selected the winter SST from December of the previous year to
January of the current year to analyze the respective decadal SST anomalies. The
results indicate that during wet years, SST has the negative ENSO phase pattern,
while in dry years, it has the positive ENSO phase pattern (Fig. 7a, b). The
reconstructed scPDSI also has the same 2–5 year cycle as ENSO (Fig. 5), suggesting
that ENSO may have contributed to drought in the Upper Heilongjiang (Amur) River
Basin. The wettest decade and the driest decade from 1950 to 2020 were also selected
for climatological analysis, which revealed the following relationships. During the
wet years, the SST in the preceding winter had the negative ENSO phase pattern, the
SST in the eastern equatorial Pacific decreased, and the western Pacific warm pool
and the Walker circulation intensified. At the same time, the western Pacific
subtropical high pressure weakened and shifted northward, the Mongolian high
pressure weakened significantly (Fig. 8a), the anomalous cyclone in the wet years
corresponded to a cold anomaly (Fig. 8c), and the major rainfall band in May–July
(MJJ) shifted northward. This scenario caused an anomalous increase in precipitation
in the Upper Heilongjiang (Amur) River Basin during the selected wet years. In dry
years, the SST in the preceding winter had an ENSO positive phase pattern, the SST
difference between the western and eastern equatorial Pacific decreased, the



latitudinal Walker circulation weakened, the western Pacific subtropical high pressure
strengthens and shifted southward compared to normal. These events result in weak
East Asian summer winds and a significantly more intense Mongolian high (Fig. 8b).
The anomalous cyclone in dry years corresponds to a warm anomaly (Fig. 8d), and
the anticyclone corresponds to a warm anomaly (Fig. 8d), which is controlled by an
eccentric northerly component that favors cold air transport from high latitudes to the
northeast during dry years. This results in anomalous descending motion and a
southward shift of the main rain and wind belts, leading to drought (Fig. 8f).
The geopotential height distance level field results show a similar pattern to that of
the Silk Road remote correlation model (Enomoto et al. 2003), which is strongly
correlated with precipitation in East Asia. The distribution of drought and
precipitation anomalies analyzed by the Silk Road remote correlation model is
consistent, suggesting that the summer drought in NEC in summer is strongly related
to the precipitation deficit. At the same time, ENSO may intensify the reduced
precipitation in NEC via its influence on the Indian summer winds, as indicated by the
Silk Road remote correlation model (Dai, 2011; Wu et al. 2003). In summary, the
large-scale ocean-atmosphere-land circulation system is a critical driver of drought
development in the Upper Heilongjiang (Amur) River Basin.
**4.3 Atmospheric water cycle during drought years**
Based on NCEP-NCAR reanalysis 1 data (Kalnay et al. 1996), we quantified the
meteorological conditions and atmospheric hydrological cycle anomalies in the Hailar
region during May–July of the driest decade of 1950–2020, based on the



reconstructed scPDSI. The total climatic precipitation for May–July of 1950–2020
was $27.0 \times 10^6$ kg/s, while the total precipitation for May–July in a drought year was
$23.0 \times 10^6$ kg/s, a decrease of 14.8%. The external advective input ($F_{in}$) under
climatic conditions was $230.9 \times 10^6$ kg/s, compared to $211.4 \times 10^6$ kg/s during the dry
year, with an 8.4% reduction in external advective input during the drought.
Evaporation ($E$) was $30.7 \times 10^6$ kg/s under these climatic conditions, and $29.5 \times 10^6$
kg/s during dry years, with a 3.9% reduction in evaporation during the drought.
Precipitation formed by external advective input ($P_a$) under these climatic conditions
was $25.3 \times 10^6$ kg/s, contributing 93.8% to precipitation, and precipitation formed by
evaporation ($P_e$) was $1.7 \times 10^6$ kg/s, with a precipitation recirculation rate of 6.2%.
Precipitation formed by external advection input ($P_a$) during the dry year was $21.4 \times$
$10^6$ kg/s, contributing 93.5% to precipitation, and precipitation formed by evaporation
($P_e$) was $1.5 \times 10^6$ kg/s, with the precipitation recirculation rate of 6.5% (Fig. 9b).
During the dry year, total precipitation decreased by 14.8% compared to the climatic
mean, and the external advective input of water vapor decreased significantly (8.4%),
resulting in a 15.4% decrease in precipitation formed from the external advective
input of water vapor, with little change in evaporation and precipitation formed by
evaporation. These results suggest that the drought in the Upper Heilongjiang (Amur)
Basin is mainly caused by a reduction in the external advective water vapor input
rather than by anomalies in the precipitation cycle. Synthetic anomalies in the whole
layer water vapor fluxes and precipitation rates also indicate a decrease in advective
water vapor transport and precipitation during the drought (Fig. 9a). These results



suggest that water vapor transport processes play a key role in the development of
drought in the Upper Heilongjiang (Amur) River Basin.
**5. Conclusion**
We built a composite tree-ring chronology for two sampling sites in the Hailar
region. Based on this chronology, we reconstructed the monthly mean scPDSI for
May–July in the Upper Heilongjiang (Amur) Basin since 1796. the reconstructed
sequence comprises more than 220 years of wet and dry variations in the Upper
Heilongjiang (Amur) River Basin, which experienced four consecutive dry periods
and five consecutive wet periods,    since 1796 CE, with a significant 2-8-year
cyclicity. The drought reconstruction accurately captured the recent trends in dry/wet
variability and it reflects drought variability across a large area.
Our synoptic climatological analysis of extreme drought years suggests that the
dry/wet variability in the Upper Heilongjiang (Amur) River Basin is related to several
large-scale climate stresses and atmospheric circulation patterns (the ENSO and Silk
Road models), and that one of the critical drivers of drought development in the
Upper Heilongjiang (Amur) River Basin is the large-scale ocean-atmosphere-land
circulation system. Our atmospheric water circulation analysis suggests that the cause
of drought is primarily a reduction in advective water vapor transport, rather than
precipitation circulation processes, which further implies that atmospheric circulation
systems control wet/dry variability in the Upper Heilongjiang (Amur) River Basin.
Our drought reconstruction has several shortcomings since it is based on only two
sample sites, and it spans a relatively short interval (230 years), and represents only a





very small region. Therefore, it is essential to systematically compile additional tree
ring–based climate records from this region to provide drought reconstructions on a
large spatial scale, which may help characterize the spatio-temporal variability and
impact mechanisms of drought within NEC.
**6. Code and data availability**
ScPDSI reconstruction in the Upper Heilongjiang (Amur) River Basin will be
available in the Supplement. The data that support the findings of this study are
available from the corresponding author upon reasonable request.
**7. Author contribution**
Feng Chen conceived the study, Yang Xu conducted the analyses and wrote the
manuscript, other authors were involved in the sample collection. All authors
interpreted and discussed the results.
**8. Acknowledgements**
This research was supported by the National Natural Science Foundation of China

(32061123008).

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



**Table**

**Table 1.** Information about the tree-ring sampling sites in the Upper Amur (Heilongjiang) River Basin.

| Site code | Lat. (N) | long. (E) | Elevation (m) | Sample | Species |
|-----------|----------|-----------|---------------|--------|---------|
| **MGET** | 121°49′ | 46°42′ | 1120 | 63/33 | *Pinus sylvestris* |
| **NEGC** | 118°44′ | 49°12′ | 1540 | 40/20 | *Pinus sylvestris* |
| **RC** | | | | 103/53 | *Pinus sylvestris* |

**Table 2.** Statistical properties of the tree-ring width chronologies from the Upper Amur (Heilongjiang) River Bas

| Statistic | MGET | NEGC | RC |
|-----------|------|------|-----|
| Mean sensitivity | 0.285 | 0.367 | 0.307 |
| Standard deviation | 0.198 | 0.21 | 0.19 |
| Mean correlation between the trees | 0.658 | 0.723 | 0.653 |
| Signal to noise ratio (SNR) | 86.651 | 60.15 | 26.063 |
| Variance of the first eigenvector (%) | 58.6 | 66.4 | 38.6 |
| First year when EPS > 0.85 (tree number) | 1762(5) | 1900(4) | 1796(10) |

**Table 3.** Results of verification and calibration tests for the scPDSI reconstruction.

| Statistical procedure | Calibration (1951-1985) | Verification (1986-2020) | Calibration (1986-2020) | Verification (1951-1985) | Full calibration (1951–2020) |
|-----------------------|---------------------------|----------------------------|---------------------------|----------------------------|------------------------------|
| R | 0.727 | 0.611 | 0.661 | 0.611 | 0.645 |
| r2 | 0.529 | 0.374 | 0.436 | 0.374 | 0.416 |
| RE | | 0.357 | | 0.491 | |
| CE | | 0.378 | | 0.566 | |
| Sign test | | 24+/11- | | 23+/12- | |
| First-order sign test | | 22+/12- | | 22+/12- | |






**Table 4.** Comparisons between the reconstructed scPDSI and documented climatic
events.

| Year | PDSI$_{5-7}$ | Local historical documents |
|---|---|---|
| 1779 | -2.93 | Famine in Taiyuan and Baotou |
| 1837 | -2.31 | Drought in Qiqihaer |
| 1842 | -2.62 | Drought in Baotou |
| 1857 | -2.28 | Drought in Baotou and the Qingshuihe river |
| 1866 | -2.79 | Drought in Hulunbuir |
| 1951 | -3.01 | Inner Mongolia region drought, decrease I in Hulunbuir grain production |
| 1996 | -2.23 | Drought in North Central Inner Mongolia in early summer |


**Figures**

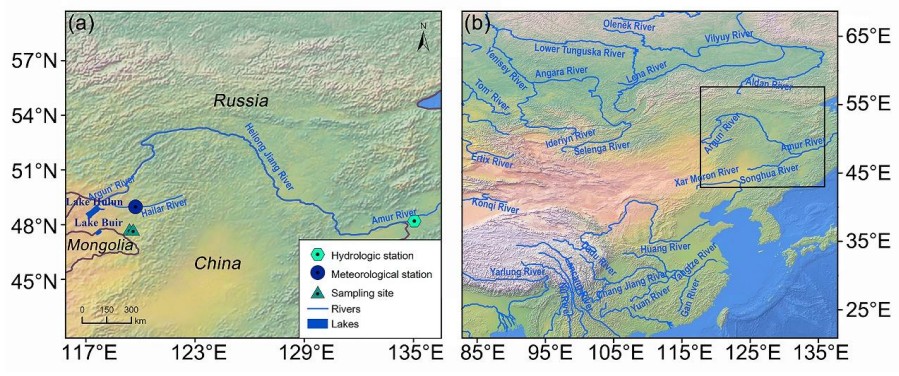


**Figure 1.** (a) Location of the tree-ring sampling sites, and meteorological and
hydrological stations in the Upper Amur (Heilongjiang) River Basin. (b) Location of
the study area in Asia. (The raster data for the production of the map was taken from
https://www.naturalearthdata.com/)

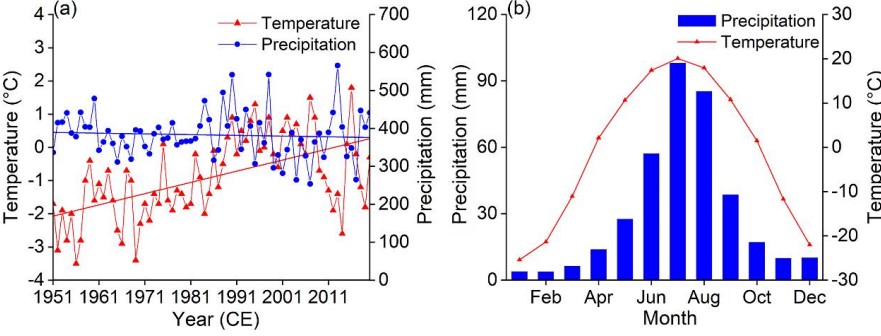




**Figure 2.** (a) Annual precipitation and temperature trends for the Upper Amur (Heilongjiang) River Basin from 1951 to 2020. (b) Monthly total precipitation and mean temperature for the Upper Amur (Heilongjiang) River Basin.

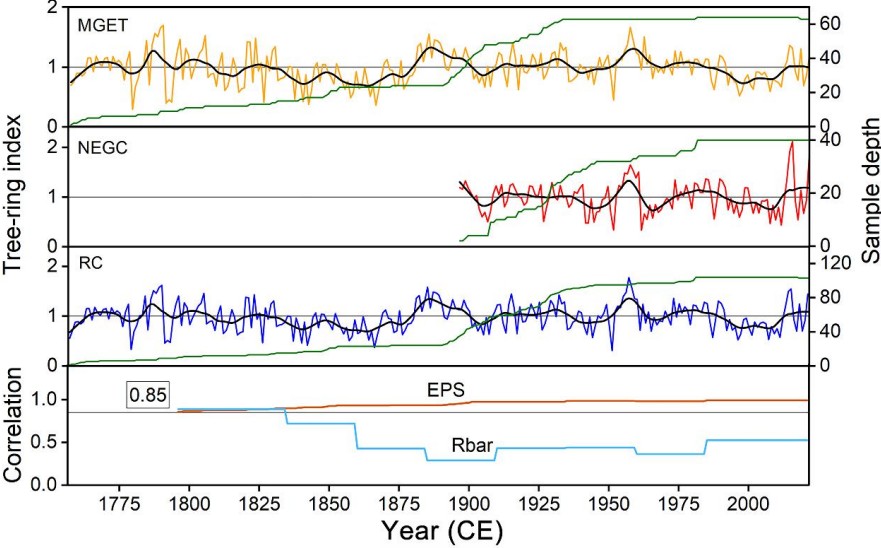

**Figure 3.** Chronologies of the two tree-rings records (MGET and NEGC) and the RC from the Upper Amur (Heilongjiang) River Basin. The inter-series correlation (Rbar) and the EPS are shown in the lowermost panel.

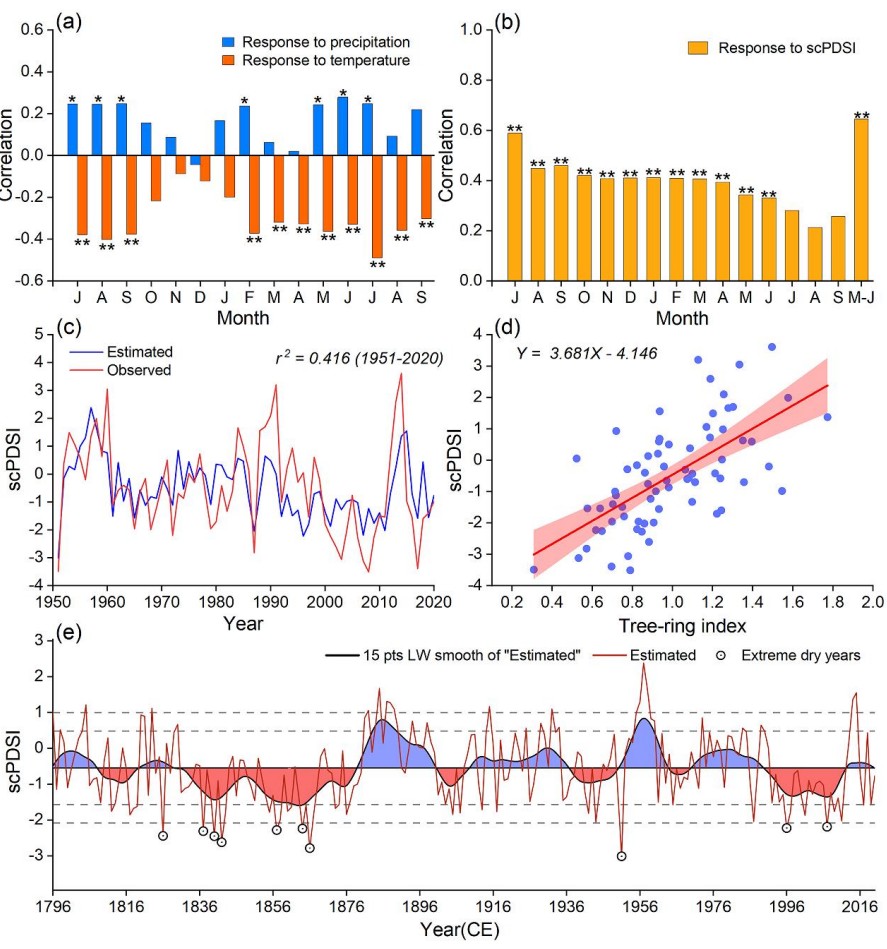

**Figure 4.** (a) Correlation coefficients between the tree-ring chronologies and monthly total precipitation and mean temperature. (b) Correlation coefficients between the RC tree-ring chronologies and monthly mean scPDSI of the CRU. Correlations are calculated from the previous June to the current September over the time period of 1951–2020 (* represent the 95% significance level, and ** represents the 99% significance level). (c) Comparison between the instrumental and reconstructed mean May–July scPDSI for the Hailar region during 1951–2020. (d) One-dimensional linear regression fits for the May to July scPDSI for 1796–2020. (e) Reconstructed mean May–July scPDSI and its 15-year low-pass filtered version since 1796 CE. The horizontal central line represents the average reconstructed scPDSI. The horizontal dotted lines represent ±1 SD and ±1.5 SD on a mean value basis.

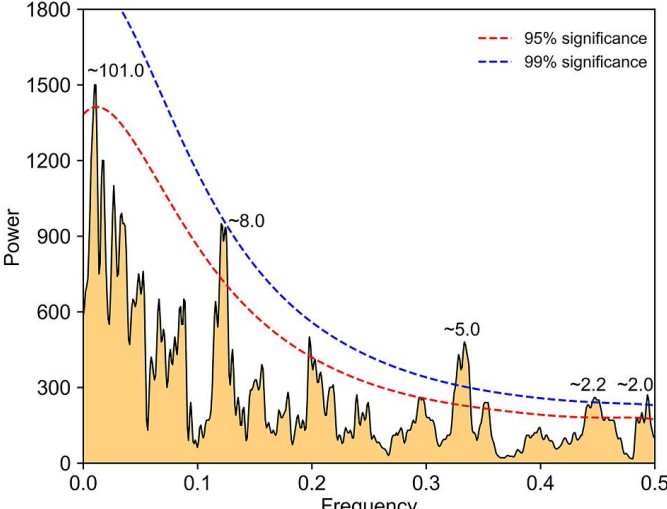

**Figure 5.** MTM spectral density of the drought reconstruction. The dashed curves represent the 95% (red) and 99% (blue) significance levels, respectively.

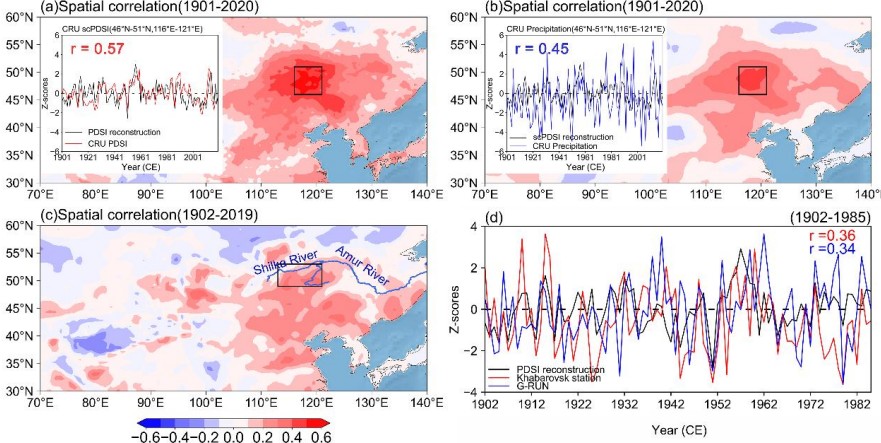

**Figure 6.** Spatial correlation maps of the reconstructed scPDSI with the CRU gridded mean May–July scPDSI (a) and the CRU gridded total May–July precipitation (b) since 1901 CE. The rectangle indicates the location of the range of the grid, and the same below. The inset graphs show a comparison of the reconstructed scPDSI with the regional mean scPDSI and precipitation curves from the CRU. (c) Reconstructed scPDSI with G-RUN gridded May-July mean runoff spatial correlation maps for the period of 1902–2019. (d) Comparison of reconstructed scPDSI, hydrological station runoff data, and the G-RUN regional mean runoff data for the period of 1902–1985.

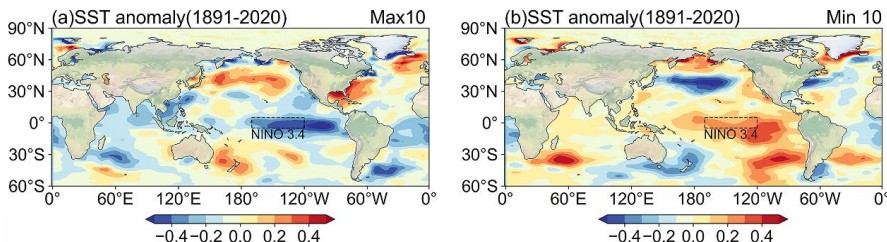

**Figure 7.** Composite maps of SST anomalies (℃) for the 10 wettest years (a) and 10
driest years (b) from the previous December to the current January during 1891–2020.

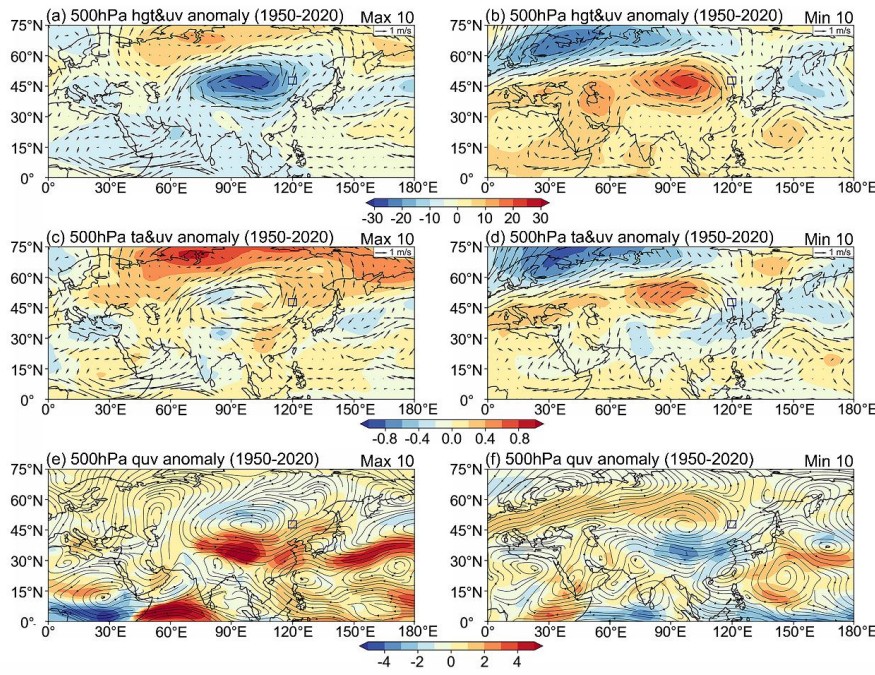

**Figure 8.** Spatial patterns of geopotential height and 500 hPa vector wind anomalies
(a, b), 500 hPa air temperature, and 500 hPa vector wind anomalies (c, d), 500 hPa
water vapor transport anomalies (e, f) in the wettest decade and the driest decade
during 1950–2020 in NCEP-NCAR Reanalysis 1. The rectangle indicates the location
of the study area.



667

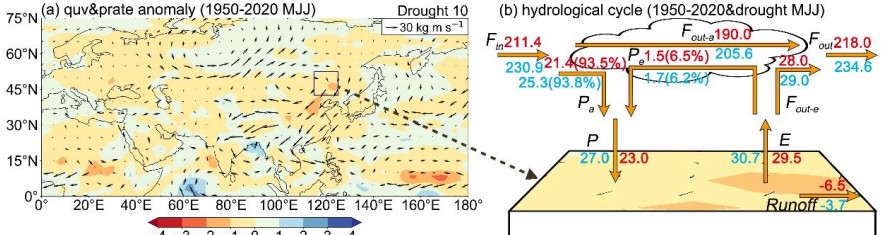

668

**Figure 9.** (a) Anomaly composites of the mean precipitation rate (kg/s·m²) and the whole layer moisture flux (kg·m/s) for May–July of the driest decade in the study area (115–125°E, 42.5–52.5°N) relative to that of May-July for the period of 1950–2020 (arrows represent the the whole layer moisture flux, filled colors represent the precipitation rate). (b) Schematic diagram of the land-atmosphere water balance in the study area during the climatic period (1950-2020) and dry years. The variables in this plot (i.e., *Fin, Fout-a, Fout-e, Fout, Pa, Pe, P, E*) are explained in Section 2.4. The blue labels (in kg/s) indicate climatic averages, while the red labels indicate averages during drought.