# Peer review of "Drought reconstruction since 1796 CE based on tree-ring widths in the"

_Climate of the Past, 2023_

## Author Comment (AC1)

Dear editors and reviewers:

On behalf of my co-authors, we thank you very much for giving us an opportunity to revise and modify our manuscript, we appreciate you and reviewers for their constructive comments and suggestions on our manuscript entitled "Drought reconstruction since 1796 CE based on tree-ring widths in the Upper Heilongjiang (Amur) River Basin in Northeast Asia, and its linkage to Pacific Ocean climate variability" (MS No. cp-2023-28).

We have studied reviewer's comments carefully and have made revision and modification. Please find below a detailed responses to all the raised points, which we would like to submit for your kind consideration.We would like to express our great appreciation to you and reviewers for comments on our paper. Looking forward to hearing from you.Thank you and best regards.

Yours sincerely,

Feng Chen

On behalf of all authors

**Comments from the Editors and Reviewers:**

**Reviewer#1:**

Manuscript title: Drought reconstruction since 1796 CE based on tree-ring widths in the Upper Heilongjiang (Amur) River Basin in Northeast Asia, and its linkage to Pacific Ocean climate variability

This manuscript presented a reconstruction of the May-July scPDSI sequence since 1796 from tree rings in the Upper Heilongjiang (Amur) River Basin. The study area is located in the climate-sensitive arid-semi-arid border region of China, which is also the location of cross-border rivers. The reconstruction is reliable and capture acceptable variance of the instrumental precipitation. The authors seem to want to link global climate change to regional extreme droughts. However, some of the analyses are preliminary and not perfect. In general, the presented results are reasonable, but the following concerns should be addressed before it can be reconsidered for publication.

**Response**: Thank you for your comments.

1) As the reconstructions in the manuscript reflect only a relatively small area of the upper Heilongjiang (Amur) River basin, in the citation section I would like the authors to indicate what the implications of your study are for other areas, please improve on this point.

**Response**: Over the past two decades, several studies based on tree-ring width have been conducted in Northeast Asia (Cook et al. 2010; Liang et al. 2007; Bao et al. 2015; Chen et al. 2012; Liu et al. 2016; Liu et al. 2009). However, research attention needs to be directed to the agro-pastoral zone in the western part of northeast China, where

the fragile ecology and climate sensitivity necessitate a greater understanding of the patterns and mechanisms of drought. The goal of our research is to find the characteristics and causes of the drought in this region.

2) In the Methods or the supplemental, it would be good to know more about how the tree-ring data were standardized and truncated to control for signal strength. Were short series removed?

**Response**: In Section 2.3, we have explained how the tree-ring data were standardized and truncated to control for signal strength. In addition, no series were removed due to short lengths in the chronologies one or several of the authors of this manuscript have produced.

3) In section 2.3, it is necessary to clarify how the correlation coefficient is calculated, as the widely used Pearson method should check whether the data set shows a normal distribution. In this case, Spearman's rank test is often used as an alternative method of calculation.

**Response**: In this study we used Pearson's correlation method, all the datasets for which the calculation of correlation coefficients was carried out were subjected to the Shapiro-Wilk normality test, and the data were tested for normality with a significance greater than 0.05, which allows for direct Pearson's correlation analysis. The table below shows some of the test results

| Shapiro-Wilk test | Tree-ring index | ScPDSI$_{5-7}$ | scPDSI$_5$ | scPDSI$_6$ | ScPDSI$_7$ | Preci$_5$ |
|---|---|---|---|---|---|---|
| statistic | 0.988 | 0.983 | 0.988 | 0.984 | 0.987 | 0.986 |
| p-value | 0.753 | 0.470 | 0.757 | 0.501 | 0.660 | 0.628 |

| Shapiro-Wilk test | Preci6 | Preci7 | Temp5 | Temp6 | Temp7 |
|---|---|---|---|---|---|
| statistic | 0.984 | 0.988 | 0.975 | 0.945 | 0.960 |
| p-value | 0.533 | 0.747 | 0.248 | 0.073 | 0.089 |

4) "2.4 Land-atmosphere water balance": This section is overloaded with descriptions and explanations of calculation methods, which should not be the focus of the methods section.

**Response:** Thank you for your reminder. We have appropriately revised some of the content to make it more abbreviated.

5) In the discussion section, the manuscript lacks a comparison with the results of reconstructions in other neighbouring areas.

**Response:** Although we agree with the concern, there are few comparable reconstructions in the neighborhood and they are relatively far away, so we chose to

validate our reconstructions by comparing them with historical documents, taking into account the effects of different climatic environments.

6) The drought index used in this manuscript is the self-calibrating Palmer Drought Severity Index (scPDSI). However, the various drought indices do not have exactly the same focus. Will the reconstructed precipitation have a similar response pattern to other drought indices (e.g. SPEI)?

**Response:** I think your confusion is that we chose the scPDSI index for the reconstruction and not the other drought indices. Due to the better physical representation of scPDSI, it has been a popular index to assess the severity of historical droughts (Chen et al. 2019) and to project future droughts from outputs of general circulation models (GCMs) (Gizaw and Gan 2016). Compared to SPEI, SPAEI and scPDSI can better represent meteorological, hydrological, and agricultural droughts of China, especially in non-humid regions (Zhang, Gan et al. 2022).

7) The hydrological data used in this study is too far from the tree ring sampling sites. It also includes the input of other branch rivers, besides the human activities.

**Response:** No, The water of hydrological station mainly comes from our sampling area, and there is no runoff producing in low elevation mountains. The area we sampled is located in the main stream of the Heilongjiang (Amur) River. This is also the main reason for the low correlation coefficient, and in the future we will add more sampling sites in the Heilongjiang (Amur) River.

8) Title of 4.1 is to discuss "Climate–tree ring growth relationships and temporal variations in regional drought". "NEC is a major food-producing region in China, and thus it is of both regional and national importance to improve our understanding of the causes and patterns of drought events and to develop appropriate responses." These sentences are not relevant to the topic of the section.

**Response:** Yes, I have corrected.

9) The year should be expressed in a unified way, some with "CE" and some without "CE".

**Response:** Yes, I have corrected.

10) Line 59: "several studies based on tree-ring width have been conducted in Northeast Asia ",What is the scientific problem in the current researches? Why did you carry out this study? Just to make a new reconstruction?

**Response:** Thank you for your comment. In recent studies, the scientific issues concerned mainly include revealing the long-term regional climate and hydrological changes and the relationship with large-scale circulation. The purposes of our study are as follows: (1) To reconstruct the scPDSI of the Upper Heilongjiang (Amur) River Basin and to analyze changes in the temporal variations of regional drought; (2) to determine the atmospheric circulation mechanisms generating extreme drought events; and (3) to analyze the contribution of advective water vapor transport and local

evaporation to precipitation during droughts, and to determine their leading causes.

11) Line 61-64: Please rephrase to make the description more clear.
**Response:** Yes, I have corrected.

12) Line 71,125: Please correct the spacing between words
**Response:** Yes, I have corrected.

13) Line 98: Show the specific meaning of "NEGC" and "MGET".
**Response:** The identification codes of these two sampling points are abbreviated from the respective Chinese place names, solely serving as designators for the sampling locations without carrying any additional connotations. Specifically, "NEGC" corresponds to Nuo'er Ga Cha, and "MGET" corresponds to Ming Ge Er Tu.

14) Line 98: The selection of the sampling sites is not sufficiently described. Lack of data about slope inclination and exposure.
**Response:** Yes, I have corrected. The missing information has been incorporated into Table 1.

15) Line 102-105:"This region has a continental and monsoonal climate. Due to the incursion of high-latitude cold and dry air masses in winter and of warm and moist air masses from low-latitude areas in summer, the climate tends to alternate between cold and dry in winter and warm and humid in summer." The description of the regional climate in the manuscript lacks an appropriate literature base.
**Response:** Yes, I have corrected.

16) Why the STD chronology was chosen among the three, the authors didn't provide the necessary explanation of the characteristics of the other two categories.
**Response:** Thank you for your comment. The tree-ring width chronology was developed with a bi-weight robust mean method using the ARSTAN program (Cook, 1985). To remove undesirable growth trends that were linked to stand dynamics and age, we chose the negative exponential functions to standardize the data. Following this method, we obtained three types of chronologies: standard (STD), residual (RES) and arstan (ARS) chronologies. Since the standard chronology (STD) yielded the strongest correlation with instrumental May-July scPDSI, we used the STD chronology for our final reconstruction. The Standard (STD) and Arstan (ARS) chronologies can retain all frequencies, the former using averaged indices and the latter by modeling the communal low frequency and adding it back to the residual (RES) chronology. The RES chronology comprises residual indices after prewhitening the STD chronology to remove low-frequency variation. The STD and ARS are usually very similar, but the ARS chronology can reduce the effects of competition in closed canopy forests (Cook, 1985).

17) Line 144,148,212,225.250,253,272: Change "correlation" to "correlation

coefficient".
**Response:** Yes, I have corrected.

18) Line 237-241: How did you get the 5 continuous wet periods and 4 continuous dry periods? What is the criterion. If you defined them by the cumulative anomaly curve , then the cumulative anomaly curve should be introduced before the periods.
**Response:** Thank you for your comment. We have a description of the definition of continuous periods in Section 2.3. To remove the interannual signal and obtain the long-term chronological signal, we smoothed our precipitation reconstruction through the low-pass filtering method for 15 years. The wet (dry) periods were identified if the smoothed values were higher (lower) than the overall mean with a break for exceeding 9 years ($\geqslant$9 years) (Li et al., 2022).

19) Line 281-300: The authors identify seven extreme drought years in the reconstruction results in the historical literature. However, the literature cited is sparse and does not provide sufficient proof of the reliability and authenticity of the results.
**Response:** Thank you for your comment. Due to the scarcity of accessible historical documents prior to 1950, we are unable to locate additional references for validation. However, this does not imply their unreliability, as these documents have been extensively utilized to substantiate historical records of droughts. Furthermore, we have also incorporated some more recent literature for corroboration.

20) Line 290-298: This sentence is tediously long, please revise it.
**Response:** Yes, I have corrected.

21) Line 307: The impact of ENSO is shown on the SST anomaly composite graph for the extreme drought years of the SST. However, as seen in Figure 7, ENSO is the most obvious, but not the only one. Do SST variations in other regions also have an impact?
**Response:** Thank you for your comment. Indeed, the sea surface temperature anomalies in the Indian Ocean also exhibit a pattern consistent with the Indian Ocean Dipole's (IOD) reversed variations. However, since our study region is situated in Northeast Asia and primarily influenced by changes in Pacific Ocean temperatures, our primary focus is on discussing the impact of ENSO (El Niño-Southern Oscillation).

22) Line 366-368: "These results suggest that water vapor transport processes play a key role in the development of drought in the Upper Heilongjiang (Amur) River Basin." This sentence should be revised.
**Response:** Yes, I have corrected.

23) Figure 1: The method of drawing Figure 1 should be given. Based on what data and software?
**Response:** We have provided explanations within the captions of the figures, The

raster data for the production of the map was taken from https://www.natura learthdata.com/. Additionally, we employed ArcGIS 10.2 for the creation of this figure.

24) Figure 3:What's the thick black curve in each figure?
**Response:** The thick black curve illustrates the 15-year low-pass filtered curve of the tree-ring width index. We have revised the figure caption to include an explanation.

25) Figure 4: High-frequency (1st order difference) comparison between the instrumental and reconstructed scPDSI should be given.
**Response:** Thank you for your reminder. We have added the correlation coefficient of the first-order difference to Section 3.1, which is 0.571. The high-frequency (1st order difference) comparison diagram is as follows.

[Figure]

26) Figure 4 (e): The curve in the diagram is not clear and I would like the author to change the colour to make it more legible.
**Response:** Yes, I have corrected.

27) Figure 6: I would like the authors to add the results of the significance test to the correlation analysis.
**Response:** Yes, I have corrected.

28) Figure 6: The small plots used as correlation between sequences in (a) and (b) are not very clear, and it is also recommended that some smoothing of the curves be added to this type of plot.
**Response:** Yes, I have corrected.

[Figure]

29) Figure 6: For spatial correlation patterns, how about the result from the meteorological station (with CRU gridded precipitation, SST)?

**Response:** Thank you for your reminder. As shown in the figure below, we calculated the spatial correlation of the May-July gridded drought index from 1951 to 2020 for comparison. As shown in the figure below, we calculated the spatial correlation between the May-July station precipitation(temperature) and CRU gridded data from 1951 to 2020. Obviously, Hailar meteorological station has significant regional dry and wet change signals, covering most areas of the Upper Heilongjiang (Amur) River Basin. The spatial correlation between station data and SST is not significant. This may be due to the short instrumental period and the dominant internal variability in regional climate change. For small regional climate change, it is difficult to connect with global climate change only through short-term correlation.

[Figure]

30) Figure 7,8,9: These 3 figures are missing units of data.

**Response:** Yes, I have corrected.

**REFERENCES**

Bao, G., Liu, Y., Liu, N., and Linderholm, H. W.: Drought variability in eastern Mongolian Plateau and its linkages to the large-scale climate forcing, Climate Dynamics, 44, 717-733, https://doi.org/ 10.1007/s00382-014-2273-7, 2015.

Chen S, Gan TY, Tan X, Shao D, Zhu J (2019) Assessment of CFSR, ERA-Interim, JRA-55, MERRA-2, NCEP-2 reanalysis data for drought analysis over China. Clim Dyn 53(1–2):737–757. https://doi.org/10.1007/s00382-018-04611-1

Chen, Z. J., Zhang, X. L., Cui, M. X., He, X. Y., Ding, W. H., and Peng, J. J.: Tree-ring based precipitation reconstruction for the forest-steppe ecotone in northern Inner Mongolia, China and its linkages to the Pacific Ocean variability, Global and Planetary Change, 86-87, 45-56, https://doi.org/10.1016/ j.gloplacha.2012.01.009, 2012.

Cook, E. R.: A time series analysis approach to tree ring standardization, University of Arizona Tucson, 1985.

Cook, E. R., Anchukaitis, K. J., Buckley, B. M., D'Arrigo, R. D., Jacoby, G. C., and Wright, W. E.: Asian Monsoon Failure and Megadrought During the Last Millennium, Science, 328, 486-489, https://doi.org/10.1126/science.1185188, 2010.

Gizaw MS, Gan TY (2016) Impact of climate change and El Niño episodes on droughts in sub-Saharan Africa. Clim Dyn 49(1–2):665–682. https://doi.org/10.1007/s00382-016-3366-2

Li, Q., Deng, Y., Wang, S.J., Gao, L.L., Gou, X.H., 2022. A half-millennium perspective on recent drying in the eastern Chinese Loess Platea. Catena. 212, 106087. https://doi.org/10.1016/j.catena.2022.106087.

Liang, E. Y., Shao, X. M., Liu, H. Y., and Dieter, E.: Tree-ring based PDSI

reconstruction since AD 1842 in the ortindag sand land, east inner mongolia, Chinese Science Bulletin, 52, 2715-2721, https://doi.org/10.1007/s11434-007-03 51-5, 2007.

Liu, N., Liu, Y., Bao, G., Bao, M., Wang, Y. C., Zhang, L. Z., Ge, Y. X., Bao, W., and Tian, H.: Drought reconstruction in eastern Hulun Buir steppe, China and its linkages to the sea surface temperatures in the Pacific Ocean, Journal of Asian Earth Sciences, 115, 298-307, https://doi.org/10.1016/ j.jseaes.2015.10.009, 2016.

Liu, Y., Bao, G., Song, H. M., Cai, Q. F., and Sun, J. Y.: Precipitation reconstruction from Hailar pine (Pinus sylvestris var. mongolica) tree rings in the Hailar region, Inner Mongolia, China back to 1865 AD, Palaeogeography Palaeoclimatology Palaeoecology, 282, 81-87, https://doi.org/10.1016/ j.palaeo.2009.08.012, 2009.

Zhang, G. X., et al. (2022). "Twenty-first century drought analysis across China under climate change." Climate Dynamics 59(5-6): 1665-1685.

**Reviewer#2:**

**Summary:** the authors have reconstructed the drought variability in the Upper Heilongjiang River Basin in Northeast Asia since 1796 based on tree-ring width and have explored its linkages with Pacific ocean-climate variability. Overall, the paper is well-written and well-organized, providing valuable insights into regional hydroclimate dynamics and mechanisms. However, there are some issues that need to be addressed.

I have one major concern:

The discussion section discusses large-scale climate driving factors that influence drought variability in the study area, including ENSO and the Silk Road teleconnection pattern, as well as the contributions of external moisture transport and local evaporation to precipitation in the atmospheric water cycle process. The discussion is insightful and plausible, but some arguments need more evidence and references to support them. For example, how do ENSO and the Silk Road teleconnection pattern affect atmospheric circulation and moisture transport over Northeast Asia? How reliable are the moisture flux data and calculations used in this study? How consistent are this study's findings with other studies in terms of climate mechanisms and impacts?

**Response:** Thank you for your comments. As we discussed in section 4.2 Synoptic meteorological analysis, El Niño events are associated with drought in Northeast Asia and La Niña events are associated with increased precipitation in Northeast Asia. The atmospheric circulation and moisture transport processes associated with them have been analysed by us. For the impact of the Silk Road teleconnection pattern, we believe that it may have had an impact on the drought in Northeast Asia because of the close link between it and the ENSO event (Zhang and Zhou 2015). However, we did not validate it in a relevant way, so we decided to remove this part of the formulation in order to make our analysis more convincing. We use the NCEP-NCAR reanalysis 1 data (Kalnay et al. 1996) to calculate moisture flux, it is widely used in atmospheric circulation analyses in East Asia (Annamalai, Slingo et al. 1999, Wu, Kinter et al. 2005, Bao, Liu et al. 2012, Song, Ke et al. 2016). The consistency of the findings of this study with those of other studies in terms of climate mechanisms and impacts is added and modified in the article.

1) In the title, the authors use "Northeast Asia", which is inconsistent with the description in the introduction (e.g., Line 44, "northeast China"). It would be better to keep it consistent.

**Response:** Yes, I have corrected.

2) In lines 60-61, overall, the cited papers seem outdated (before 2018). I suggest citing more recent papers (since 2019) from nearby regions.

**Response:** Yes, I have corrected.

3) Line 82: What does "EU" stand for? It is unclear.

**Response:** It means "European Union". I have corrected.

4) Line 91: Revise "To reconstruct" to "to reconstruct".

**Response:** Yes, I have corrected.

5) Line 116: Revise "chest height" to "breast height".
**Response:** Yes, I have corrected.

6) In lines 215-222: May-July scPDSI was found to be the dominant climatic signal in tree-ring widths. How about linkages of tree-ring width with SPEI at different scales? Maybe higher correlations could be detected.
**Response:** We used CSIC SPEI data (Begueria et al. 2010) for correlation coefficients and found that the correlation coefficients between the SPEI and the tree-rotor width index were much lower than the scPDSI, which did not satisfy the reconstruction requirements. The calculation results are presented in the following figure.

[Figure]

7) Lines 234: Please specify what is meant by low-frequency. Which time domain? Inter-decadal?
**Response:** Thank you for your reminder. Yes, the low-frequency represent the inter-decadal signals in the reconstructed scPDSI, and there may be some problems with the presentation here. I have corrected.

8) Lines 262-263: Usually, the full Latin name is used for the first time, and abbreviations of genus names are shown for the rest. Therefore, please revise "Pinus sylvestris var. Mongolica" to "P. sylvestris var. Mongolica".
**Response:** Yes, I have corrected.

9) Lines 591-593: In Table 1, the species name is "Pinus sylvestris", different from "Pinus sylvestris var. Mongolica" in the body text. Please keep consistent.
**Response:** Yes, I have corrected.

10) Lines 642-650: Add the Amur River in (a) and (b), as shown in (c).
**Response:** Yes, I have corrected.

**REFERENCES**

Annamalai, H., et al. (1999). "The mean evolution and variability of the Asian summer monsoon: Comparison of ECMWF and NCEP-NCAR reanalyses." Monthly Weather Review **127**(6): 1157-1186.

Bao, G., et al. (2012). "April-September mean maximum temperature inferred from Hailar pine (Pinus sylvestris var. mongolica) tree rings in the Hulunbuir region, Inner Mongolia, back to 1868 AD." Palaeogeography Palaeoclimatology Palaeoecology **313**: 162-172.

Begueria, S., et al. (2010). "A MULTISCALAR GLOBAL DROUGHT DATASET: THE SPEIBASE A New Gridded Product for the Analysis of Drought Variability and Impacts." Bulletin of the American Meteorological Society **91**(10): 1351-1354.

Song, C. Q., et al. (2016). "Homogenization of surface temperature data in High Mountain Asia through comparison of reanalysis data and station observations." International Journal of Climatology **36**(3): 1088-1101.

Wu, R. G., et al. (2005). "Discrepancy of interdecadal changes in the Asian region among the NCEP-NCAR reanalysis, objective analyses, and observations." Journal of Climate **18**(15): 3048-3067

Zhang, L. X. and T. J. Zhou (2015). "Drought over East Asia: A Review." Journal of Climate **28**(8): 3375-3399.